# The Efficacy of Instrumental Physical Therapy through Extracorporeal Shock Wave Therapy in the Treatment of Plantar Fasciitis: An Umbrella Review

Francesco Agostini [1,*], Massimiliano Mangone [1], Nikolaos Finamore [1], Marta Di Nicola [2], Federico Papa [3], Giuliano Alessio [1], Luigi Vetrugno [4], Angelo Chiaramonte [1], Giorgia Cimbri [1], Andrea Bernetti [1], Marco Paoloni [1] and Teresa Paolucci [5]

[1] Department of Anatomical and Histological Sciences, Legal Medicine and Orthopedics, Sapienza University, Piazzale Aldo Moro 5, 00185 Rome, Italy; massimiliano.mangone@uniroma1.it (M.M.); nikolaosfinamore@yahoo.it (N.F.); giuliano.alessio@uniroma1.it (G.A.); angelo.chiaramonte@uinroma1.it (A.C.); giorgia.cimbri@uniroma1.it (G.C.); andrea.bernetti@uniroma1.it (A.B.); marco.paoloni@uniroma1.it (M.P.)

[2] Laboratory of Biostatistics, G. D'Annunzio University of Chieti-Pescara, Via dei Vestini, 31, 66100 Chieti, Italy; marta.dinicola@unich.it

[3] Physiotherapy, G. D'Annunzio University of Chieti-Pescara, Via dei Vestini, 31, 66100 Chieti, Italy; federico.papa@unich.it

[4] Department of Anesthesia, Critical Care and Pain Medicine, SS. Annunziata Hospital, G. D'Annunzio University of Chieti-Pescara, Via dei Vestini, 31, 66100 Chieti, Italy; luigi.vetrugno@unich.it

[5] Unit of Physical and Rehabilitation Medicine, Department of Medical, Oral and Biotechnological Sciences, G. D'Annunzio University of Chieti-Pescara, Via dei Vestini, 31, 66100 Chieti, Italy; teresa.paolucci@unich.it

\* Correspondence: francescoagostini.ff@gmail.com; Tel.: +39-340-475-1090

**Abstract:** (1) Background: Plantar fasciitis (PF) is the most common cause of heel pain in adults. Extracorporeal shockwave therapy (ESWT) is a minimally invasive treatments commonly used for treating PF. Our aim is to provide a complete overview of which treatments have been compared to ESWT, with a focus on the modalities that have been used. (2) Methods: A thorough search of the literature was performed on Medline via Pubmed, Cochrane Database of Systematic Reviews (CDSR) of the Cochrane Library and Physiotherapy Evidence Databases (PEDro) up to 18 November 2021. In the study were included only systematic reviews and meta-analysis in English language, published from 2010 to date. (3) Results: A total of 14 systematic reviews and meta-analysis were included in the umbrella review. A total of eight studies compared the efficacy of ESWT treatment with placebo, three studies compared ESWT with another therapy (two studies compared ESWT and corticosteroids, one study ESWT and ultrasound therapy), and three studies had more than one comparison. (4) Conclusions: When compared to placebo, ESWT demonstrated to be effective. More randomized trials with specific comparisons between different types and intensity of SW are needed to obtain more precise information on SW effectiveness.

**Keywords:** extracorporeal shock wave therapy; foot pain; plantar fasciitis; rehabilitation

## 1. Introduction

Plantar fasciitis (PF) is the most common cause of heel pain in adults [1,2]. Inflammation is classically considered the main pathogenic mechanism, but no evidence about inflammation has been found in most studies. Otherwise, evidence of degenerative changes in the plantar fascia led many authors to identify it as "fasciosis" [3]. PF usually generates pain that involves the medial calcaneal tuberosity and is often worse in the morning [1]. For the majority of patients, symptoms are self-limiting [4–6]. Conservative treatments are considered the first approach: they consist of rest, nonsteroidal anti-inflammatory drugs (NSAIDs) or foot orthotics. About 85% to 90% of patients do not need to undergo surgery,

and 80% of patient do not experience pain relapse after conservative treatments [7–10]. Minimally invasive treatments commonly used for treating PF are extracorporeal shockwave therapy (ESWT), corticosteroid (CS) injections, platelet-rich plasma (PRP) injections, botulinum toxin (BTX), acupuncture, dry needling and prolotherapy [2,11–13]. ESWT is a physical therapy that generates three-dimensional pressure pulses, lasting microseconds and reaching peek pressures of 35–120 MPa, and has effects depending on intensity, pulse cycle and shockwave (SW) modality [14–16]. We have two modalities of SW therapy: focused shockwave (FSW) and radial shockwave (RSW). FSW is documented as a possible alternative to the surgical approach: it focuses on a small area (2 to 8 mm) and penetrates at a selected depth, having effects that depend on the energy delivered to the focal area; that is why it is important to know the energy flux density (EFD), which is considered the "dose" of SW administered [17,18].

RSW produces SW that are transmitted radially and do not have penetrating effects on tissue, acting superficially. They are frequently used in soft-tissue pathologies and, recently, also in plantar fasciitis [19]. As this physical therapy is widely used for the treatment of plantar fasciitis [20,21], a wide number of studies have arisen to analyze its efficacy, comparing ESWT with other different treatments (especially with placebo or corticosteroids), but these are often head-to-head comparisons with another therapy [22,23]. Furthermore, little attention has been paid to the setting parameters of this physical therapy. The aim of our umbrella review is to provide a complete overview of the effects of ESWT compared to placebo or other treatments, with a focus on the ESWT modalities used. This might be useful to all the authors trying to test new therapies and for clinicians to choose the best therapy available to them. To date, to our knowledge, only one umbrella systematic review has been conducted on the argument [24], generally analyzing the epidemiology, evaluation, and treatment of PF. Our review is specifically focused on the evaluation of SW's effects on treating PF: it analyzes the effects on pain, the types of SW, the energy levels administered and complications.

## 2. Materials and Methods

A thorough search of the literature was performed on Medline via PubMed, Cochrane Database of Systematic Reviews (CDSR) of the Cochrane Library and Physiotherapy Evidence Databases (PEDro) up to November 2021. In the study were included only systematic reviews and meta-analysis in the English language, published from 2010 to date. A specific research protocol was not registered before the study's start. To perform this search, we used an association of the Mesh terms "plantar fasciitis" and "shock wave", "radial extracorporeal shockwave therapy" ("RSWT"), "focal shockwave therapy" ("FSWT") or "extracorporeal shock wave therapy" (ESWT) connected with different Boolean operators (research terms are reported in Supplementary Paper S1). We then screened our results searching for systematic reviews and meta-analyses. Two operators (F.A. and A.B.) independently screened and reviewed all the studies found by the search. Full text articles were obtained when the article was considered important, and duplicated articles were excluded. Disagreement and inconsistencies were overcome by discussion between reviewers and consulting a third reviewer (M.P.). We included systematic reviews and meta-analysis about the use of extracorporeal shock wave therapy (ESWT) in patients diagnosed for plantar fasciitis, so defined by the trial author, regardless of the timing of the symptom's arousal. We included only studies focusing on humans, which enrolled adult people (≥18 years old). We considered as interventions ESWT treatment (focused and radial extracorporeal shockwave therapy), compared with other physical/pharmacological therapies or with placebo. We considered as outcomes of interest the following: treatment success and pain relief, analyzed with numerical or other specific rating scores. We also screened the bibliography of the studies included, searching for more systematic reviews or meta-analyses following the inclusion criteria. We excluded articles that were not possible to obtain, articles without an abstract, and articles that considered PF together with other painful foot pathologies. We also excluded articles when ESWT was associated with other physi-

cal or pharmacological therapies. The Preferred Reporting Items for Systematic Reviews and Meta-analysis (PRISMA) statement guidelines were followed to write this umbrella review [25]. Articles eligible for inclusion were assessed for methodological quality using the AMSTAR checklist [26,27].

## 3. Results

A total of 16 systematic reviews and meta-analyses was included in the umbrella review [11,14,20,21,28–37]. Our primary search led to 38 articles. We removed duplicates, and after a first screening of titles and abstract only, 13 articles were considered relevant to our scope and were included. Three systematic reviews [28,38,39] obtained by the bibliographies of the selected reviews were considered relevant and added to the study, for a total of sixteen articles (selection progress is shown in Figure 1). All results are available and synthesized in Table 1.

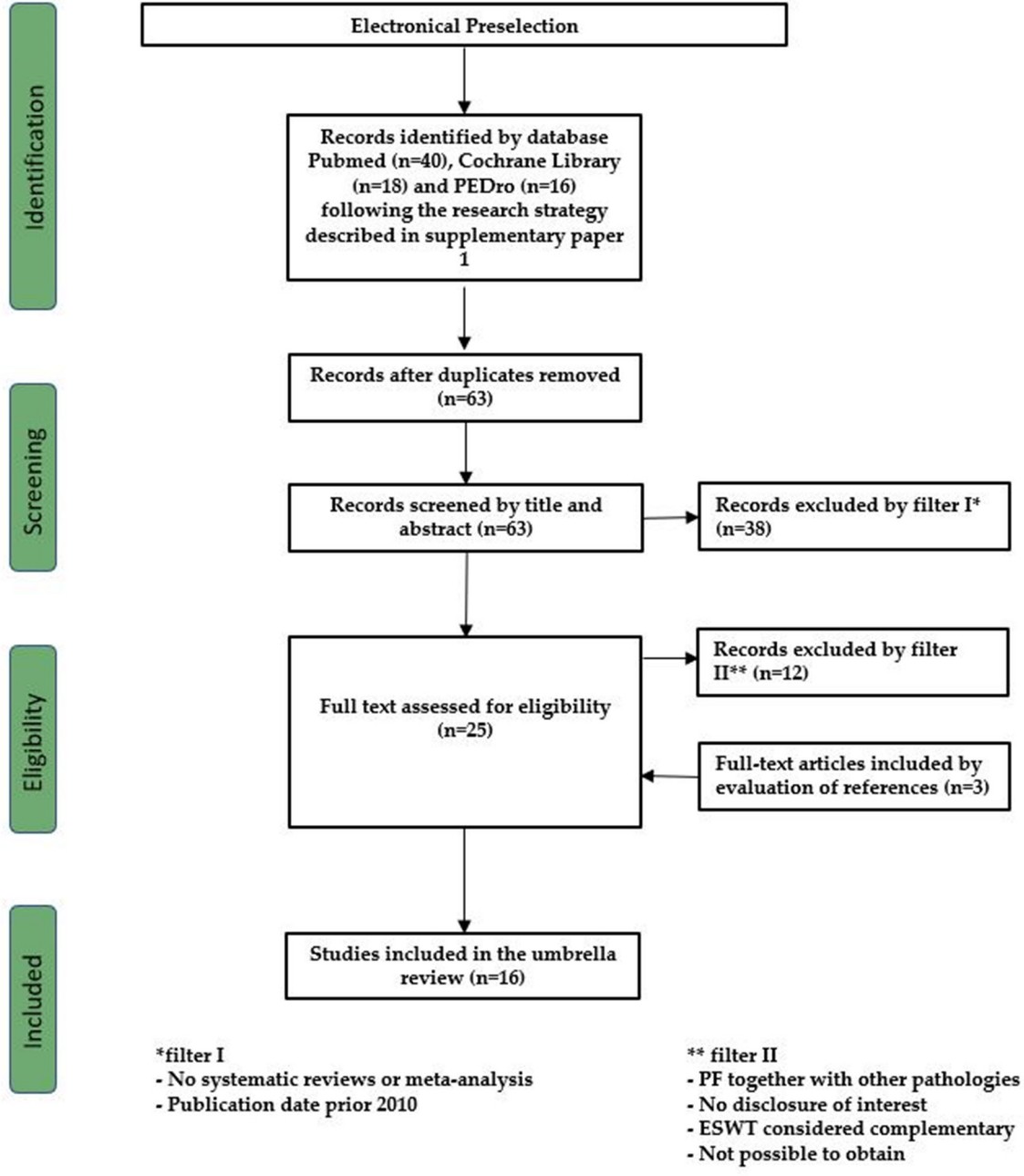

**Figure 1.** Flow chart.

**Table 1.** Included study.

| Study | Participants | Outcome | Intervention/Control | | Results | Conclusions |
|---|---|---|---|---|---|---|
| Chang, K.V. et al., 2012 [20] | Participants (n = 1431) between 25 and 87 years old Patients complaining of heel pain near the proximal plantar fascia on the medial calcaneal tuberosity. Symptoms that lasted for more than 3 months. | Success rates (defined as pain reduction more than 50/60% than baseline) Reduction in pain: VAS score | FSW divided in: (1) low intensity (EFD < 0.08 mJ/mm$^2$), (2) medium intensity (EFD 0.08–0.28 mJ/mm$^2$), (3) high intensity (EFD > 0.28 mJ/mm$^2$), RSW therapy as another group | Placebo | Twelve articles included The Jadad scale was used for validity assessment Trials with scores less than 3 were considered to have lower methodologic quality and were not selected for further meta-analysis. | The meta-analysis supports the use of SW therapy for plantar fasciitis. Success rates were not related to energy levels, pain reduction might disclose a slight dose–response relationship. FSW: Authors suggest the highest EFD in the range of medium intensity. RSW: is recommended for its lower price and likely better effectiveness. |
| Dizon, J.N. et al., 2013 [29] | Participants (n = 1287), ages from 18 to 79 years, all clinically diagnosed to have chronic heel pain | Pain reduction: overall pain, morning pain, activity pain: VAS score Functional outcome: RM score Adverse effects | Low energy to high-energy ESWT (1) low intensity, <0.1 mJ/mm$^2$; (2) moderate intensity, 0.1–0.2 mJ/mm$^2$; (3) high intensity, >0.2 mJ/mm$^2$ | Placebo Standard Treatment options | Articles included: - 11 to analyze adverse effects. - 8 for effectiveness (pain reduction and functional outcomes). PEDro scale used to analyze methodological quality: all considered strong. | ESWT, using moderate and high intensity, are effective in reducing pain and improving function. |
| Aqil, A. et al., 2013 [30] | Participants (n = 663) Patients with PF not responding to a minimum of 3 months of nonoperative treatment | Pain reduction; morning pain; pain during activity: all with VAS score Success rate: RM score | ESWT without local anesthetic:<br>• Three studies used FSW.<br>• Three studies used RSW.<br>• One study did not specify | Placebo | Seven RCTs included The quality was rated using the Scottish Intercollegiate Guidelines Network scoring system and the methods described by Jadad et al. | In patients with PF not responsive to conservative or other nonoperative measures for a minimum of 3 months, ESWT without a local anesthetic is more effective than with a placebo. |
| Zhiyun, L. et al., 2013 [35] | Participants (n = 716), aged over 18 years, suffering from recalcitrant PF: pain from over 6 months and conservative treatment failure | Pain relief: VAS score | High intensity ESWT (HESWT): ESWT with energy > 0.2 mJ/mm$^2$ | Placebo | Five RCTs included Methodological quality assessed with Jadad score | HESWT is more effective than placebo on recalcitrant PF. |
| Yin, M.C. et al., 2014 [21] | Participants (n = 550) aged ≥ 18 Patients who had plantar fasciitis for >6 months | Success treatment rate and pain: VAS score Function: RM score | ESWT divided into 2 intensity levels: (1) low intensity (<0.2 mJ/mm$^2$); (2) high intensity (>0.2 mJ/mm$^2$) | Placebo Plantar fasciotomy (only 1 of the studies include) | 7 RCTs were included Methodological quality assessed using the Jadad scale | The efficacy of low-intensity ESWT is worthy of recognition. Pain relief and functional outcomes are satisfactory in the short-term. |

**Table 1.** *Cont.*

| Study | Participants | Outcome | Intervention/Control | Results | Conclusions |
|---|---|---|---|---|---|
| Hsiao, M.Y. et al., 2015 [14] | Participants (n = 604), older than 18 years, with recalcitrant PF: pain for >3 months of conservative treatment failure | Pain relief: VAS score. OR of treatment success rate | ESWT (FSW and RSW considered together): 2 or 3 treatment sessions, EFD from 0.02 to 0.42 mJ/mm$^2$ | CSs ABPs (autologous blood-derived products) | Seven RCT and 3 quasi-experimental studies Methodological quality assessed with Jadad scale for RCT, Newcastle-Ottawa scale for quasi-experimental studies | In the short-term follow up (3 months), ABPs has the best results, followed by CSs. At 6 months, shockwaves and ABPs have better results than CSs. |
| Sun, J. et al., 2017 [31] | Participants (n = 935) Patients suffering from heel pain and diagnosed for PF | Success rate (reduction in VAS > 50–60% than baseline) Pain reduction (VAS score) Complications | General ESWT (comprising both FSW and RSW). FSW. RSW | Placebo | Nine studies included (6 FSW vs. placebo, 3 RSW vs. placebo) Methodological quality was assessed with Cochrane Risk of bias tool | FSW seems to have higher success rate and greater pain reduction than placebo. No solid conclusion can be drawn on general ESWT and RSW. |
| Lou, J. et al., 2017 [36] | Patients > 18 years old with recalcitrant PF | Pain evaluation: overall pain; morning pain; pain during activity; RM score | ESWT without any conservative treatment or local anesthetic: not specified neither if FSW or RSW nor intensity levels | Placebo | Nine RCTs included Risk of bias assessed with Cochrane Handbook Systematic Review of Interventions | ESWT seems to be effective in relieving pain in patients with PF. |
| Sun, K. et al., 2018 [28] | Participants (n = 1185) Patients suffering from heel pain and clinically diagnosed for PF | Success rates, Reduction of pain, Return to work time, Complications, Function (RM score) | ESWT without local anesthetic | Placebo | Thirteen articles included Methodological quality assessed with Cochrane's Handbook 5.1.0. | ESWT had better results on RM score, reduction in pain scales, return to work time, success rate. |
| Xiong, Y. et al., 2018 [32] | Participants (n = 454) older than 18 years | Pain and functional subscales: VAS score, 100 Scoring System, Mayo CSS, FFI, HTI | SW therapy: not specified if FSW or RSW; different intensity levels and protocol used | CSs (not unique protocol of administration) | Six articles included in the meta-analysis Methodological quality and risk of bias were assessed with modified Jadad scale and Cochrane Handbook for Reviews of Interventions | Both SW and CSs are effective on pain relief and self-reported function improvement, with not significant inter-group differences (SW had better results of pain). |
| Li, H. et al., 2018 [33] | Participants (n = 177) older than 18 years | Pain and functional outcomes: VAS score; AOFAS; PFPS; FFI | ESWT: not specified if RSW or FSW. Different intensities and protocols | Ultrasound therapy (UT), with different protocols | Five trials included in the meta-analysis Methodological quality and risk of bias assessed with modified Jadad scale and Cochrane Handbook for Reviews of Interventions | Both ESWT and UT are effective in relieving pain and improving self-reported function. SW has better results, but no significant differences are found between groups. |

**Table 1.** *Cont.*

| Study | Participants | Outcome | Intervention/Control | Results | Conclusions |
|---|---|---|---|---|---|
| Li, S. et al., 2018 [34] | Participants (n = 658) Patients with PF and without injection history | Pain reduction: VAS score. Success rate (VAS decrease >50% than baseline) and recurrence rate. Functional outcomes: FFI, Mayo CSS, AOFAS. Adverse events | ESWT, not specified if RSW or FSW. Two intensity levels: (1) Low intensity (<0.2 mJ/mm$^2$) (2) High intensity (>0.2 mJ/mm$^2$) | Ultrasound-guided CSs injection | Nine RCTs included Risk of bias assessed with Cochrane risk of bias tool | Pain relief and success rate are related to intensity level: high-intensity ESWT has the best results, followed by CSs and low-intensity ESWT. |
| Li, H. et al., 2018 [37] | Participants (n = 2889) with PF | Pain relief: VAS score Overall efficacy | (1) Nonsteroidal anti-inflammatory drugs (NSAIDs). (2) CSs. (3) Botulinum Toxin A (BTX-A); (4) Dry Needling (5) Autologous whole blood. (6) Platelet-rich Plasma (PRP) (7) Ultrasound Therapy (8) ESWT: no distinction in FSW and RSW; protocol not specified | | Forty-one RCT Methodological quality assessed with Jadad scale | ESWT has the best results at three- and six months follow-up and is judged the most effective treatment. |
| Li, X. et al., 2018 [11] | Participants (n = 1676) older than 18 years diagnosed with plantar fasciitis | Pain relief: VAS score, the pain subscale of FFI (Foot Function Index) | (1) Low-level Laser Therapy (LLLT), (2) Ultrasound Therapy (UT), (3) Intracorporeal Pneumatic shock therapy (IPST), (4) Ultrasound-guided pulsed radiofrequency (UG-PRF) (5) Non-invasive interactive neurostimulation (NIN) (6) ESWT: FSW divided in three intensity levels following Chang [13] classification; RSW considered separately | The studies included in the meta-analysis evaluated at least 2 treatment modalities, including sham therapy | Nineteen articles were included in the meta-analysis Methodological quality assessed with Cochrane collaboration Risk of bias tool | RSW seems to be more effective and to have more stable effects on pain relief. UT and FSW therapies can be considered treatment candidates. UG-PRF and high intensity FSW are not recommended. More studies are needed for NIN, UG-PRF, IPST and LLT. |
| Babatunde, O.O. et al., 2018 [39] | Participants (n = 2450). Adults with PHP (PF, plantar fasciopathy, plantar fasciosis) | Pain and functional disability | (1) ESWT: no distinction made by intensity or generator (2) Corticosteroid injections (3) NSAIDs (4) Orthoses (5) Exercise | | Thirty-one RCTs included. Risk of bias assessed with Cochrane Collaboration's Risk of Bias tool | No treatment significantly better than others in short- (<6 weeks), medium- (6–12 weeks) and long-term (>12 weeks) follow-up. |

**Table 1.** *Cont.*

| Study | Participants | Outcome | Intervention/Control | Results | Conclusions |
|---|---|---|---|---|---|
| Wang, J.C. et al., 2019 [38] | Participants (n = 1714). Adults with PF | Pain relief: VAS score; Treatment success rate | ESWT divided into three intensity levels: low (EFD < 0.1 mJ/mm$^2$); medium (0.1–0.2 mJ/mm$^2$); high ($\geq$0.2 mJ/mm$^2$). Distinction between RSW and FSW | Placebo | Fourteen RCT included. Risk of bias assessed with Cochrane Handbook Systematic Review of Interventions | Medium-energy ESWT is more effective up to 12 months follow up compared to placebo. Efficacy of low- and high-energy ESWT is uncertain. |

Legenda: CSs—corticosteroids; AOFAS—American Orthopedic Foot and Ankle Society; EFD—energy flux density; FFI—Foot Functional Index; OR—odds ratio; PFPS—plantar fasciitis pain and disability scale; RM score—Roles and Maudsley score; VAS—visual analogue scale.

We excluded 11 articles because they were duplicates; it was not possible to obtain 1 of the studies; 2 articles were excluded because they were published prior to 2010; 1 was excluded because it focused only on complications of using shock wave therapy in PF; 1 article was excluded because ESWT treatment was considered as a complementary intervention, 2 were excluded because considered plantar fasciitis together with other pathologies; the other articles were excluded because they did not meet our inclusion criteria.

*Methodological Quality*

Methodological quality was assessed using the AMSTAR2 criteria [27]. Two reviewers independently valued the articles included. Discrepancies within the evaluation of single items were resolved through discussion. The kappa score for the evaluation of the interrater reliability was 0.74, with a substantial agreement between reviewers. The quality varied between the studies from critically low (14 studies out of 16, 87.5%) to low (2 studies, 12.5%), as we can see in Table 2. Methodological weaknesses in several studies often referred to three critical domains: Q2, Q7 and Q15.

A total of nine studies [20,21,28–31,35,36,38] compared the efficacy of the ESWT treatment with placebo, three studies [32–34] compared ESWT with another therapy (two studies compared ESWT and corticosteroids, one study ESWT and ultrasound therapy), and four studies [11,14,37,39] had more than one comparison.

In their systematic review and network meta-analysis, Chang, K.V. et al. [20] compared the effectiveness of FSW of different intensity levels and RSW in 1431 patients suffering from plantar fasciitis for at least 3 months. They included in their study 12 randomized clinical trials (RCTs) and performed a pairwise and network meta-analysis on the success rate of intervention and reduction in pain scales at 6 months. In both the outcomes in the study, pairwise comparisons highlighted that medium- and high-intensity FSW therapies had greater success; in the network meta-analysis, RSW had the highest effectiveness versus placebo, followed by low-, medium- and high-intensity FSW. They also investigated the relationship between intensity levels of FSW and the outcomes and found that the success of treatment was not related to intensity, though pain reduction might be dose dependent. Their conclusion is that RSW is an appropriate treatment for plantar fasciitis; if FSW has to be used, the highest and mostly tolerable energy efflux density without anesthesia in the range of medium intensity is the preferable option.

Dizon, J.N. et al. [29] performed a meta-analysis comparing ESWT and placebo in patients diagnosed with PF with a 6 months or longer duration of pain. The treatment with ESWT had different intensities, from low (<0.1 mJ/mm$^2$) to moderate (0.1–0.2 mJ/mm$^2$) and high (>0.2 mJ/mm$^2$). The outcomes of the study, assessed at 12 weeks after intervention, were the visual analogue scale (VAS), pain (during the first steps in the morning and during activity) and improvement in functional outcome using the Roles and Maudsley score.

The study involved 1287 patients and concluded that moderate- and high-intensity ESWT treatments are effective in reducing pain and improving function in patients with chronic plantar fasciitis.

**Table 2.** Methodological quality of included studies.

| | Q1 | Q2 | Q3 | Q4 | Q5 | Q6 | Q7 | Q8 | Q9 | Q10 | Q11 | Q12 | Q13 | Q14 | Q15 | Q16 | Overall Assessment |
|---|---|---|---|---|---|---|---|---|---|---|---|---|---|---|---|---|---|
| Chang, K.V. et al., 2012 [20] | Y | N | Y | PY | Y | Y | N | Y | PY | N | Y | Y | Y | Y | N | Y | CRITICALLY LOW |
| Dizon, J.N. et al., 2013 [29] | Y | N | Y | PY | N | N | N | PY | Y | N | Y | N | Y | Y | N | Y | CRITICALLY LOW |
| Aqil, A. et al., 2013 [30] | Y | N | N | PY | Y | Y | N | PY | N | N | Y | N | N | Y | N | Y | CRITICALLY LOW |
| Yin, M.C. et al., 2014 [21] | Y | N | N | PY | N | Y | N | PY | Y | N | Y | Y | N | Y | N | Y | CRITICALLY LOW |
| Sun, J. et al., 2017 [31] | Y | N | N | PY | N | Y | Y | PY | Y | N | Y | Y | Y | Y | N | Y | CRITICALLY LOW |
| Li, X. et al., 2018 [11] | Y | Y | N | PY | Y | Y | PY | PY | Y | Y | Y | Y | Y | Y | N | Y | LOW |
| Xiong, Y. et al., 2018 [32] | Y | N | N | PY | Y | Y | N | PY | PY | N | Y | N | Y | Y | N | Y | CRITICALLY LOW |
| Li, H. et al., 2018 [33] | Y | N | N | PY | Y | Y | N | PY | Y | N | Y | Y | Y | Y | N | Y | CRITICALLY LOW |
| Li, S. et al., 2018 [34] | Y | N | N | PY | Y | Y | N | PY | Y | N | Y | Y | N | Y | Y | Y | CRITICALLY LOW |
| Zhiyun, L. et al., 2013 [35] | Y | N | N | PY | Y | Y | Y | PY | Y | N | Y | Y | Y | Y | N | Y | CRITICALLY LOW |
| Hsiao, M.Y. et al., 2015 [14] | Y | N | N | PY | Y | Y | N | Y | PY | N | Y | N | N | Y | Y | Y | CRITICALLY LOW |
| Li, H. et al., 2018 [37] | Y | N | N | PY | Y | N | N | Y | PY | N | Y | N | N | Y | Y | Y | CRITICALLY LOW |
| Lou, J. et al., 2017 [36] | Y | N | N | PY | Y | Y | N | PY | Y | N | Y | Y | N | Y | N | Y | CRITICALLY LOW |
| Sun, K. et al., 2018 [28] | Y | N | N | PY | Y | Y | N | PY | Y | N | Y | N | Y | N | N | Y | CRITICALLY LOW |
| Wang, J.C. et al., 2019 [38] | Y | N | N | PY | Y | Y | N | PY | Y | N | Y | Y | Y | Y | N | Y | CRITICALLY LOW |
| Babatunde, O.O. et al., 2018 [39] | Y | PY | Y | PY | Y | Y | Y | Y | Y | N | Y | Y | Y | Y | N | Y | LOW |

**Legenda**: Y = yes, N = no, PY = partial yes, N/A = non applicable. Q1: Did the research questions and inclusion criteria for the review include the components of PICO? Q2: Did the report of the review contain an explicit statement that the review methods were established prior to the conduct of the review and did the report justify any significant deviations from the protocol? Q3: Did the review authors explain their selection of the study designs for inclusion in the review? Q4: Did the review authors use a comprehensive literature search strategy? Q5: Did the review authors perform study selection in duplicate? Q6: Did the review authors perform data extraction in duplicate? Q7: Did the review authors provide a list of excluded studies and justify the exclusions? Q8: Did the review authors describe the included studies in adequate detail? Q9: Did the review authors use a satisfactory technique for assessing the risk of bias (RoB) in individual studies that were included in the review? Q10: Did the review authors report on the sources of funding for the studies included in the review? Q11: If meta-analysis was performed, did the review authors use appropriate methods for statistical combination of results? Q12: If meta-analysis was performed, did the review authors assess the potential impact of RoB in individual studies on the results of the meta-analysis or other evidence synthesis? Q13: Did the review authors account for RoB in primary studies when interpreting/discussing the results of the review? Q14: Did the review authors provide a satisfactory explanation for, and discussion of, any heterogeneity observed in the results of the review? Q15: If they performed quantitative synthesis, did the review authors carry out an adequate investigation of publication bias (small study bias) and discuss its likely impact on the results of the review? Q16: Did the review authors report any potential sources of conflict of interest, including any funding they received for conducting the review?

Aquil, A. et al. [30] in their meta-analysis compared the effectiveness of ESWT treatment with placebo on patients suffering from PF for a minimum of 3 months. The outcomes of the study were composite scores of pains, success rate of heel improvement, heel pain improvement when taking first steps, heel pain improvement while doing daily activities, heel pain improvement after application of a pressure meter and the Roles and Maudsley score. All outcomes but one had major improvement with ESWT rather than placebo. Authors conclude recommending ESWT to be used.

Yin, M.C. et al. [21] conducted a systematic review and meta-analysis of seven RCTs that compared ESWT and another treatment (placebo or plantar fasciotomy). They analyzed the success rate of the intervention and its effectiveness on pain relief and function. They found that low-intensity ESWT had better results than placebo in all the outcomes of the study, though high-intensity ESWT had poorer results.

In their meta-analysis, Sun, J. et al. [31] included nine RCTs, comparing FSW and RSW therapies with placebo in patients suffering from plantar fasciitis. They found that FSW had higher success rates and major pain relief than the placebo; despite positive results, RSW had a high heterogeneity, and no solid conclusions could be drawn on this therapeutic methodology.

In their systematic review and network meta-analysis, Li, X. et al. [11] included 19 RCTs and compared the effectiveness of different therapies for treating plantar fasciitis: ESWT (both FSW and RSW), ultrasound therapy (UT), low-level laser therapy (LLLT), intracorporeal pneumatic shock therapy (IPST), non-invasive interactive neurostimulation (NIN) and pulsed radiofrequency (PR). The main outcome on study was pain relief (an-

alyzed with VAS, the foot function index or other indices) at different follow-up points: short term (0–6 weeks), intermediate term (2–4 weeks) and medium term (6–12 months). The network meta-analysis highlighted that NIN had the highest probability of providing the best outcome at 0–6 weeks, RSW at 2–4 months and IPST at 6–12 months, but RSW had only slightly worse results than the best treatment therapy at the first and third follow-up periods. Authors indicated also that FSW was inferior to RSW at all time intervals, with low intensity being preferable to other intensities.

Xiong, Y. et al. [32] included in their meta-analysis six RCT (with 454 participants involved) on PF, and ESWT treatment was compared with corticosteroids (CS) injections. They evaluated at three months the pain reduction with the VAS in all studies, with self-reported outcome scores (Mayo CSS, FFI, HFI, and 100 scoring system score) for four of them. They found that ESWT and CS had similar results in improving self-reported functional scores and better results on relieving pain, but no statistical differences were found between groups.

Li, H. et al. [33] in their meta-analysis compared the effectiveness of ESWT and ultrasound treatments in patients with PF. They found that both ESWT and ultrasound are effective in relieving pain and improving self-reported function.

Li, S. et al. [34] in their meta-analysis analyzed pain reduction, treatment success rate and functional outcomes at three months of ESWT and CS injection treatment in patients with PF. Nine RCTs were included in the study, and the ESWT intensity levels were divided into low ($<0.2$ mJ/mm$^2$) and high ($>0.2$ mJ/mm$^2$). The authors concluded that high-intensity ESWT was superior to CS both in pain reduction and treatment success rate; low-intensity ESWT had worse results than CS injections in the same outcomes. The functional outcomes did not indicate significative differences between treatment groups. The authors also highlighted the importance of intensity levels in providing functional outcomes.

Zhiyun, L. et al. [35] performed a meta-analysis comparing high-energy extracorporeal SW therapy (HESWT) with placebo in patients with recalcitrant PF (over 6 months, conservative treatment failure). They enrolled 716 patients and analyzed as the main outcome the pain reduction with the VAS score at a 12-week follow up. They found that HESWT is an effective treatment for recalcitrant PF and recommend it after conservative care failure and before surgical intervention.

Lou, J. et al. [36] performed a meta-analysis on RCT that compared ESWT (without any other conservative treatment or local anesthetic) with placebo. The study involved 1174 patients and analyzed at 12 weeks post treatment changes in overall heel pain, heel pain upon taking the first step in the morning, overall heel pain during daily activities, improvement of Roles and Maudsley score to "excellent" or "good", and reduction in overall heel pain using an F-Meter. In all these items, ESWT showed better results than the placebo.

Hsiao, M.Y. et al. [14] performed a pairwise and network meta-analysis comparing the effectiveness of autologous blood-derived products (ABPs), CS and SW therapies. Outcomes on study were pain relief and treatment success at 3 and 6 months. They included three studies comparing SW and CS therapies, and one comparing platelet-rich plasma (PRP) and SW therapies. Radial and focused SW therapies were considered together; patients underwent two or three treatment sessions; the energy efflux intensity ranged from 0.02 to 0.42 mJ/mm$^2$. The pairwise comparisons found no significant differences in the three treatments at 6 months regarding pain reduction and treatment success. The network meta-analysis on pain relief found that at 3 months, ABPs were the best treatment and SW therapy had poorer results; at 6 months, SW therapy had the best results, slightly better than ABPs. Treatment success analysis showed that differences between the treatment modalities are small.

Li, H. et al. [37] conducted a network meta-analysis of eight different therapies (CS injections, ESWT, ultrasound therapy (US), botulinum-toxin A (BTX-A), dry needling (DN), autologous whole blood (AWB), nonsteroidal anti-inflammatory drugs (NSAIDs), and PRP), analyzed changes in the VAS at 1, 2, 3, 6 months, and ranked the results of the therapy by

utilizing the surface under cumulative ranking curve (SUCRA). They found that ESWT was more efficacious than the other seven therapies. In particular, it was superior to the placebo at the 1-, 2- and 3-month VAS change analyses and it was superior to the placebo and CSs at the 6-month analysis.

Sun, K. et al. [28] in their meta-analysis compared ESWT without anesthesia treatment and placebo in patients diagnosed with chronic plantar fasciitis. In total, 13 studies and 1185 patients were included in the meta-analysis; success or improvement rates, RM score, pain reduction, return to work time and complications were evaluated as outcomes (not specifying the follow up timing). ESWT had better results than other therapy in all the outcomes on study. The authors concluded that ESWT is more effective than placebo, which seems to have negative effects on the outcomes.

Wang, J.C. et al. [38] in their meta-analysis analyzed pain relief (at VAS score) and success rate of ESWT treatment compared with placebo at different follow-up points (1, 3, 6 and 12 months). They compared placebo with different types of SW, and made a distinction based on energy levels and the type of SW administered (FSW or RSW). They concluded that medium-energy ESWT without local anesthesia was more effective on the outcomes on study, regardless of the type of generator.

Babatunde, O.O. et al. [39] in their systematic review with a network meta-analysis compared the effectiveness of the pain and functional disability of five commonly used therapies on plantar heel pain (PHP), a term comprehensive of PF. Between the five therapies (orthoses, exercise therapy, corticosteroid injections, NSAIDs and ESWT), no statistically significant differences were found at short- (<6 weeks), medium- (6–12 weeks) and long-term (>12 weeks) follow-ups.

## 4. Discussion

Plantar fasciitis is the most common cause of heel pain in adults. Despite the pain in PF often resolving within one year regardless of treatment, conservative treatments are often the first approach: ESWT was introduced in the 1990s [4], but it has not always been considered effective [40,41]. It is important to underline that, when specified, all the studies included in the umbrella review reported at least 3 months from symptom onset [11,14,20,21,29,30,35], or the authors referred to the pathology as "chronic" [28,31]. None of the studies treat the acute pathology (symptoms from <6 weeks), which is a debated argument [42].

### 4.1. Effects on Pain

All the studies included in the review report pain reduction, due to the treatment with shockwaves. The VAS score was used to evaluate pain in the studies. Three studies [29,30,36] evaluated this score at different conditions (overall pain, morning pain, and activity pain). In the majority of cases [21,29,30,32,35,36,39], this outcome was reported at 12 weeks or 3 months. Only one study [20] considered a single follow-up period of 6 months. Four studies [11,14,34,37] analyzed the outcomes more than once at the follow-up, with the longest period being 6 months. Wang, J.C. et al. [38] extended the follow-up period to 12 months. Three studies [28,31,33] did not consider the timing of the outcomes' assessment, with different follow-up periods. ESWT treatment is widely studied, but the biomechanical mechanisms of pain reduction are still uncertain. SW might stimulate tissue regeneration and neovascularization; they modulate pain transmission by acting on the levels of substance P, destruct unmyelinated nerve fibers and also have a role on destroying calcifications in tendons [18,41]. Therapy might need up to 16 weeks to actualize these biomechanical modifications [43–45]. Therefore, this might invalidate any speculation about the long-term effects, as the outcomes were measured at short (3 months) or medium (6 months) follow-up periods. The only study with a 12-month follow-up [38] assessed the effectiveness of medium-energy ESWT.

### 4.2. Functional Outcomes

Nine studies [21,28–30,32–34,36,39] reported the effectiveness of SW therapy on functional outcomes: in particular, in five of them [21,28–30,36], the Roles and Maudsley score (RM score) was used, evaluating self-reported pain and limitation of activity. Two studies [33,34] reported different functional scores: foot functional index (FFI), American Orthopedic Foot and Ankle score (AOFAS), plantar fasciitis pain and disability scale (PFPS), and Mayo clinic scoring system score (Mayo CSS). When compared to the placebo [29,30,36], functional outcomes were reported as improved. Sun, K. [28] also reported a reduced return-to-work time in patients that underwent ESWT treatment.

### 4.3. Intensity Levels and Type of SW Administered

Shockwaves are differently classified in studies. One study [31] separated SW as radial and focused. In this study, the authors found that FSW are effective on treating PF when compared with a placebo; despite promising results, no solid conclusions were drawn on RSW, due to the heterogeneity of results, probably related to the low quality of evidence included in the study. A double distinction by intensity levels and type of wave (radial or focused) administered was applied only in two studies [11,20]. These studies do not clarify whether one type of SW is superior to the other: both authors concluded that RSW are effective (Li, X. et al. [11] concluded that RSW is the most effective treatment for PF). The authors gave different conclusions on the FSW: Chang, K.V. et al. [20] stated that medium-intensity FSW is more effective, while Li, X. et al. [11] concluded that only low-intensity FSW is effective (placebo was superior to medium- and high-intensity FSW). This result is the opposite of what Zhiyun, L. et al. [35] demonstrated with their study, stating the effectiveness of high-intensity ESWT on PF. Additionally, Wang, J.C. et al. [38] compared the effects on the pain and success rate of medium-energy RSW/FSW with a placebo: both focused and radial SW had better results than the placebo. One limit of these studies is that they did not directly compare the different types of SWs but compared the results with a placebo. The authors believe that a difference in the type and the intensity of shockwave administered should be provided in future studies to better evaluate the effects of these treatments. Moreover, the studies often compared shockwaves without a rigorous protocol, with differences in the intensity levels, number of administrations, and pulses administered; this might have invalidated the results of some of them. Five studies [11,21,29,34,35,38] analyzed the effects of SW by the intensity level. In particular, the intensity levels were differently classified: Chang, K.V. et al. [20] identified three intensity levels, low ($<0.08$ mJ/mm$^2$), medium ($0.08$ to $0.28$ mJ/mm$^2$) and high ($>0.28$ mJ/mm$^2$); Li, X. et al. [11] cited Chang following the same division; Dizon, J.N. et al. and Wang, J.C. et al. [29,38] also identified three levels, with different intensities (low $<0.1$ mJ/mm$^2$, medium $0.1–0.2$ mm$^2$ and high $> 0.2$ mm$^2$); and Li [34], Zhiyun, L. et al. [35] and Yin, M.C. et al. [21] identified two levels, low ($<0.2$ mJ/mm$^2$) and high ($>0.2$ mJ/mm$^2$). As suggested by Chang, K.V. et al. [20], to obtain a therapeutic effect, it is important to provide a sufficient energy efflux density (EFD), which might be related to the reduction in the magnitude of pain, with a potential dose-dependent response. In the studies that compared different intensities of SW, three [20,29,34] assessed a better efficacy of medium- or high-intensity ESWT treatment rather than low-intensity in PF, though two studies [11,21] reported better results with low-intensity treatment. Researchers might better explore this aspect in the near future.

### 4.4. Comparisons

When compared with placebo [20,21,28–31,35,36], ESWT demonstrated to be effective, regardless of the type of shockwave administered. Two studies compared ESWT with corticosteroids [32,34]: one of them [32] concluded that both have similar effects at three-month follow-up; the other [34] made a distinction between high- and low-intensity SW levels, and concluded that HESWT is the best treatment, followed by CS and LESWT. CS was demonstrated to be effective in treating PF; however, the effects of this treatment

usually last for a short term and CS has a recognized risk of rupture of the plantar fascia, fat pad atrophy and recurrence of symptoms [32,33]. In a study that compared SW therapy, ABP and CS [14], SW therapy was considered to have the lowest probability of being the best treatment at three months but had the same probabilities of ABP to be the best treatment at 6 months. Despite this article potentially underestimating the effectiveness of SW treatment due to the low intensity levels administered, it supports the long-term effectiveness of the therapy, with the possible biomechanical changes previously described. Changes in plantar fascia thickness or other parameters related to ESWT treatment are also visible at imaging [46,47]: the authors believe that studies analyzing radiological changes associated with functional improvements and clinical pain reduction might demonstrate the effectiveness of this treatment in the near future.

### 4.5. Complications

Adverse effects were specifically reported and analyzed in seven reviews [21,28,29,31,34–36]. These are often reported as local edema, erythema, paresthesia, bruise and pain (during and a few days after the intervention). One systematic review by Roerdink, R.L. et al. [48] specifically focused on the complications related to SW treatment: it included 39 studies and highlighted that pain during treatment was the most reported side effect (255 out of 1820 participants), more frequently reported in high-intensity rather than low-intensity treatment. Two severe adverse events (out of 2229 participants) were reported: one of skin infection and one of precordial pain. Both were judged as not likely to be related to ESWT treatment.

In our review, adverse effects were specifically compared between treatment groups in two studies [28,29]: Dizon, J.C. et al. [29] reported a major incidence of calcaneal pain and erythema in ESWT group, while Sun, K. et al. [28] reported major incidence in the control group. Lou, J. et al. [36] reported that one patient withdrew from an RCT included after a loss of consciousness due to the magnitude of pain. Li, S. et al. [34] reported four patients with severe migraine or headache in one of the included RCTs. However, no other severe adverse effects, such as infections, tendon rupture or abnormal musculoskeletal events, were reported in the studies included.

Future research should focus on the design of more trials with stronger methodological quality, assessing the effectiveness on PF of the different therapies: in particular, more attention has to be paid to the type and the parameters of the SW administered. More articles with similar and more objective outcome measures are needed to perform new meta-analysis and systematic reviews assessing the effectiveness of treatments for PF.

### 4.6. Limitations

There are some limitations in our review. First, as this is only a narrative umbrella review, no solid conclusions can be drawn on the argument. Moreover, the study protocol was not published prior to conducting the study. Secondly, our search strategy led to a small number of articles with different methodological quality that might invalidate our conclusion. We only included studies written in English, so some relevant studies in other different languages may have been missed. More articles with a stronger methodological selection process are needed; moreover, the reviews often analyzed the same RCTs.

## 5. Conclusions

The present umbrella review compared the results of 16 studies comparing ESWT with other treatments. When compared to placebo, ESWT demonstrated to be effective. Most of the studies analyzed the short- or medium-term effectiveness, and there is confusion on the parameters set for the administration of the shockwaves. In particular, more attention has to be paid to the difference between FSW and RSW and the intensity of the SW administered. More randomized trials with specific comparisons between different types and intensity of SW are needed to obtain more precise information on the effectiveness of SW.

**Supplementary Materials:** The following supporting information can be downloaded at: https://www.mdpi.com/article/10.3390/app12062841/s1, Supplementary Paper S1.

**Author Contributions:** Conceptualization, T.P. and F.A.; methodology, M.M.; software, M.P.; validation, A.B., M.D.N. and G.A.; formal analysis, N.F.; investigation, F.P.; resources, F.A.; data curation, T.P.; writing—original draft preparation, N.F.; writing—review and editing, F.A., A.C. and G.C.; visualization, A.B. and L.V.; supervision, M.P.; project administration, T.P. All authors have read and agreed to the published version of the manuscript.

**Funding:** This research received no external funding.

**Institutional Review Board Statement:** Not applicable.

**Informed Consent Statement:** Not applicable.

**Data Availability Statement:** Not applicable.

**Conflicts of Interest:** The authors declare no conflict of interest.

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
