# Peer review of "The Efficacy of Instrumental Physical Therapy through Extracorporeal Shock Wave Therapy in the Treatment of Plantar Fasciitis: An Umbrella Review"

_applsci, doi:10.3390/app12062841_

Round 1
Reviewer 1 Report
# Although this article was the only umbrella review about ESWT for plantar fasciitis, the quality of study was not as high as an umbrella review we expected.
First, the number of including meta-analysis articles was small. One of the reasons may be related to the restricted language and date of publication (from 2010 to 2018?). More meta-analysis articles should be included, especially several related studies published during 2018-2020.
Second, the results should be systemically divided into several subtitles (such as pain, functional outcomes, different mode or intensity of ESWT…) to discuss. There was no focus on results and no valid conclusion on discussion.
Third, there were several limitations, such as varied methodological quality of included articles, the different mode and intensity of ESWT in included studies, the unclear aim of this study with diverse outcome measurement of meta-analysis articles, no effective method to compare included articles.
Besides, for “materials and methods”, as following:
# LINE 71: Why chose articles in English only? How about Italian?
# LINE 71: Why restricted date of paper from 2010?
# LINE 77: Please provide the name of operators.
# LINE 80: Please provide the name of third reviewer.
# LINE 94: Please cite the latest PRISMA flowchart.
# LINE 116: How to evaluate the interrater reliability of assessing methodology quality of included articles? Kappa score can be considered, if needed.
Author Response
REVIEWER 1
Reviewer 1: Although this article was the only umbrella review about ESWT for plantar fasciitis, the quality of study was not as high as an umbrella review we expected.
Authors: Dear reviewer, thanks for the comment.
Reviewer 1: First, the number of including meta-analysis articles was small. One of the reasons may be related to the restricted language and date of publication (from 2010 to 2018?). More meta-analysis articles should be included, especially several related studies published during 2018-2020.
Authors: Dear reviewer, thanks for the comment. One of the reasons might be the restricted selection criteria, since authors considered only plantar fasciitis, though a higher number of articles could be found analyzing heel spur or a general “plantar heel pain”. We also included our search strategy as supplementary material. Our search considered the period since 2010 to date, since authors considered important to focus on the latest evidence of the literature. This period also includes the major number of reviews on the argument. We chose the English language because the international literature is written in this language.
Reviewer 1: Second, the results should be systemically divided into several subtitles (such as pain, functional outcomes, different mode or intensity of ESWT…) to discuss. There was no focus on results and no valid conclusion on discussion.
Authors: Dear reviewer, thanks for the comment. We divided our discussion into different subtitles to better discuss the outcomes on study.
Reviewer 1: Third, there were several limitations, such as varied methodological quality of included articles, the different mode and intensity of ESWT in included studies, the unclear aim of this study with diverse outcome measurement of meta-analysis articles, no effective method to compare included articles.
Authors: Dear reviewer, thanks for the comment. We are aware about the limitations, and we expanded the “limitations” section of the discussion.
Reviewer 1: Besides, for “materials and methods”, as following:
Authors: Dear reviewer, thanks for the comment.
Reviewer 1: LINE 71: Why chose articles in English only? How about Italian?
Authors: Dear reviewer, thanks for the comment. We focused on the English literature since all reviews with international impact are edited in English language.
Reviewer 1: LINE 71: Why restricted date of paper from 2010?
Authors: Dear reviewer, thanks for the comment. Our search considered the period since 2010 to date, since authors considered important to focus on the latest evidence of the literature. This period also includes the major number of reviews on the argument
Reviewer 1: LINE 77: Please provide the name of operators.
Authors: Dear reviewer, thanks for the comment. We have added the initials of the two operators.
Reviewer 1: LINE 80: Please provide the name of third reviewer.
Authors: Dear reviewer, thanks for the comment. We have modified the manuscript specifying the initials of the third reviewer.
Reviewer 1: LINE 94: Please cite the latest PRISMA flowchart.
Authors: Dear reviewer, thanks for the comment. We modified the citation with the latest PRISMA statement version.
Reviewer 1: LINE 116: How to evaluate the interrater reliability of assessing methodology quality of included articles? Kappa score can be considered, if needed.
Authors: Dear reviewer, thanks for the comment. As you suggested, we evaluated the interrater reliability with Kappa score.
Reviewer 2 Report
This is a well-written manuscript with an important clinical message, and should be of great interest to the readers . The hypothesis efficacy of instrumental physical therapy through extracorporeal shock wave therapy in the treatment of plantar fasciitis is a so is very important in order to help readers about a better knowledge.
The topic of this study is trend, so it may be helpful to the readers.
On the other hand efficacy ofxtracorporeal shock wave therapy in the treatment of plantar fasciitis could reduce the injuries rates.
Introduction section is deep enough with and adequate focus that may help readers to improve knowledge about the topic. However authors should improve the stay of art, for example including references to plantar fascitis I suggest to include this references include in the attached to complet this requeriment
doi: 10.1016/j.disamonth.2021.101210
methods section is right written
Results section is clearly showed with an enough number of figures and tables that help to achieve a better understanding of this analysis.
Discussion section is well structured with different sections. Authors manage well the discussion leading a good comparison with the showed references. However I suggest discuss their achievements with mos recently literature i.e. the outcomes of the following research
doi.org/10.5114/aoms/143122
Conclussions are supported by the shown data.
Author Response
REVIEWER 2
Reviewer 2: This is a well-written manuscript with an important clinical message and should be of great interest to the readers. The hypothesis efficacy of instrumental physical therapy through extracorporeal shock wave therapy in the treatment of plantar fasciitis is a so is very important in order to help readers about a better knowledge.
Authors: Dear Reviewer, thanks for the comment.
Reviewer 2: The topic of this study is trend, so it may be helpful to the readers. On the other hand, efficacy of extracorporeal shock wave therapy in the treatment of plantar fasciitis could reduce the injuries rates.
Authors: Dear Reviewer, thanks for the comment.
Reviewer 2: Introduction section is deep enough with and adequate focus that may help readers to improve knowledge about the topic. However, authors should improve the stay of art, for example including references to plantar fasciitis I suggest including this references include in the attached to complete this requirement. doi: 10.1016/j.disamonth.2021.101210
Authors: Dear Reviewer, thanks for the comment. We have added the reference as you suggested.
Reviewer 2: methods section is right written
Authors: Dear Reviewer, thanks for the comment.
Reviewer 2: Results section is clearly showed with an enough number of figures and tables that help to achieve a better understanding of this analysis.
Authors: Dear Reviewer, thanks for the comment.
Reviewer 2: Discussion section is well structured with different sections. Authors manage well the discussion leading a good comparison with the showed references. However, I suggest discuss their achievements with mos recently literature i.e. the outcomes of the following research doi.org/10.5114/aoms/143122
Authors: Dear Reviewer, thanks for the comment. We have added the reference as you suggested.
Reviewer 2: Conclusions are supported by the shown data.
Authors: Dear Reviewer, thanks for the comment.
Reviewer 3 Report
In summary, this is an umbrella review investigating the efficacy of shockwave therapy in the treatment of plantar fasciitis. Given that there is an already-published umbrella review (see comment below) that encompasses this topic and is more thorough, this review is not unique. At best, the authors need to significantly improve their manuscript in order for this manuscript to be considered for publication. Please see comments below.
Line 35-39: break it into 2-3 sentences.
Line 40: “they consist in” à they consist of
Line 42: “from 85 to 90%” à “about 85 to 90%”
Line 43: don’t à do not (please do not abbreviate don’t, that’s, didn’t etc); a comma is needed before “and”
I will not comment on grammatical and punctuation errors from now on. Authors should double check.
Line 65-66: Rhim et al. has already published “A systematic review of systematic reviews on the epidemiology, evaluation, and treatment of plantar fasciitis” that already covers the contents of this review. I do not find this review unique. I would not phrase “this is the first umbrella systematic review” unless acknowledging the preexisting review by Rhim et al.
Line 71: Why from 2010 to date? Is there good rationale?
Line 71-72: Why didn’t authors publish the protocol? This is an essential step before starting on any type of systematic reviews these days.
Line 73-75: Please provide full search terms and corresponding results as a supplement
Line 95: Why didn’t’ authors use ARMSTAR2, the most updated version?
Line 101-102: Authors seem to miss several systematic reviews.
-For example, Sun 2020 Extracorporeal shock wave therapy versus other therapeutic methods for chronic plantar fasciitis, Wang 2019 Efficacy of different energy levels used in focused and radial extracorporeal shockwave therapy in the treatment of plantar fasciitis, Babatunde 2019 Comparative effectiveness of treatment options for plantar heel pain
-Also, there is a systematic review on complications that the authors can consider: Roerdink 2017 Complications of extracorporeal shockwave therapy in plantar fasciitis
-These are just few examples and I wonder if the authors did the thorough search. I recommend reading the umbrella review written by Rhim et al. I mentioned above because this could potentially help authors identify more studies to include.
-Line 116, 120. Please comment on the methodological quality. Also, recommend redoing it based on ARMSTAR2.
-Discussion: Discussion needs to be better organized. Currently, there are different concepts packed into one huge paragraph. Can be divided into several paragraphs dealing with efficacy, different energy level, complications, limitations, etc.
-Line 258: there is more including: Wang 2019 Efficacy of different energy levels used in focused and radial extracorporeal shockwave therapy in the treatment of plantar fasciitis
Author Response
REVIEWER 3
Reviewer 3: In summary, this is an umbrella review investigating the efficacy of shockwave therapy in the treatment of plantar fasciitis. Given that there is an already-published umbrella review (see comment below) that encompasses this topic and is more thorough, this review is not unique. At best, the authors need to significantly improve their manuscript in order for this manuscript to be considered for publication. Please see comments below.
Authors: Dear Reviewer, thanks for the comment.
Reviewer 3: Line 35-39: break it into 2-3 sentences.
Authors: Dear Reviewer, thanks for the comment. We modified the manuscript as you suggested.
Reviewer 3: Line 40: “they consist in” à they consist of
Authors: Dear Reviewer, thanks for the comment. We modified the manuscript as you suggested.
Reviewer 3: Line 42: “from 85 to 90%” à “about 85 to 90%”
Authors: Dear Reviewer, thanks for the comment. We modified the manuscript as you suggested.
Reviewer 3: Line 43: don’t à do not (please do not abbreviate don’t, that’s, didn’t etc); a comma is needed before “and”
Authors: Dear Reviewer, thanks for the comment. We modified the manuscript as you suggested.
Reviewer 3: I will not comment on grammatical and punctuation errors from now on. Authors should double check.
Authors: Dear Reviewer, thanks for the comment. All the grammatical and punctuation errors found were modified.
Reviewer 3: Line 65-66: Rhim et al. has already published “A systematic review of systematic reviews on the epidemiology, evaluation, and treatment of plantar fasciitis” that already covers the contents of this review. I do not find this review unique. I would not phrase “this is the first umbrella systematic review” unless acknowledging the preexisting review by Rhim et al.
Authors: Dear Reviewer, thanks for the comment. We were not aware of the article by Rhim et al., as it was published online on 24 November 2021 and our search was concluded at first days of November. We changed our manuscript, and we included this study in our review, to better discuss an argument of great interest nowadays.
Reviewer 3: Line 71: Why from 2010 to date? Is there good rationale?
Authors: Dear Reviewer, thanks for the comment. Our search considered the period since 2010 to date, since authors considered important to focus on the latest evidence of the literature. This period also includes the major number of reviews on the argument.
Reviewer 3: Line 71-72: Why didn’t authors publish the protocol? This is an essential step before starting on any type of systematic reviews these days.
Authors: Dear Reviewer, thanks for the comment. We included our search strategy as a supplementary PDF.
Reviewer 3: Line 73-75: Please provide full search terms and corresponding results as a supplement
Authors: Dear Reviewer, thanks for the comment. As you requested, we included our search terms as a supplementary PDF to the manuscript.
Reviewer 3: Line 95: Why didn’t’ authors use ARMSTAR2, the most updated version?
Authors: Dear Reviewer, thanks for the comment. As you suggested, we modified the manuscript and evaluated the methodological quality of the studies included in the review with AMSTAR2.
Reviewer 3: Line 101-102: Authors seem to miss several systematic reviews.
Authors: Dear Reviewer, thanks for the comment. We added the studies you suggested to our review and we are confident to have overcome this problem.
Reviewer 3: -For example, Sun 2020 Extracorporeal shock wave therapy versus other therapeutic methods for chronic plantar fasciitis, Wang 2019 Efficacy of different energy levels used in focused and radial extracorporeal shockwave therapy in the treatment of plantar fasciitis, Babatunde 2019 Comparative effectiveness of treatment options for plantar heel pain
Authors: Dear Reviewer, thanks for the comment. The first study you suggested (Sun 2020 Extracorporeal shock wave therapy versus other therapeutic methods for chronic plantar fasciitis) has been already analyzed in our review, as it refers to an article accepted in November, 2018 and published in 2020. We also added the other two articles you suggested to our review.
Reviewer 3: -Also, there is a systematic review on complications that the authors can consider: Roerdink 2017 Complications of extracorporeal shockwave therapy in plantar fasciitis
Authors: Dear Reviewer, thanks for the comment. We included the study you suggested in “Complications” section of the discussion.
Reviewer 3: -These are just few examples and I wonder if the authors did the thorough search. I recommend reading the umbrella review written by Rhim et al. I mentioned above because this could potentially help authors identify more studies to include.
Authors: Dear Reviewer, thanks for the comment. We read the umbrella review by Rhim et al. and included the studies you suggested.
Reviewer 3: -Line 116, 120. Please comment on the methodological quality. Also, recommend redoing it based on ARMSTAR2.
Authors: Dear Reviewer, thanks for the comment. As you suggested, we used the AMSTAR2 to evaluate the methodological quality and we added comments on the methodological quality.
Reviewer 3: -Discussion: Discussion needs to be better organized. Currently, there are different concepts packed into one huge paragraph. Can be divided into several paragraphs dealing with efficacy, different energy level, complications, limitations, etc.
Authors: Dear Reviewer, thanks for the comment. We divided in paragraphs and re-organized the discussion as you suggested.
Reviewer 3: -Line 258: there is more including: Wang 2019 Efficacy of different energy levels used in focused and radial extracorporeal shockwave therapy in the treatment of plantar fasciitis
Authors: Dear Reviewer, thanks for the comment. We included the study by Wang 2019 to the results and the discussion.
Round 2
Reviewer 1 Report
# The included study in table 1 can be listed in order of published year.
Author Response
REVIEWER 1
Reviewer 1: The included study in table 1 can be listed in order of published year.
Authors: Dear reviewer, thanks for the comment. We modified the Table 1 and ordered the studies by publication date.
Reviewer 2 Report
autors have adresed all my requeriment in the correct way
Author Response
REVIEWER 2
Reviewer 2: Authors have adresed all my requeriment in the correct way
Authors: Dear Reviewer, thanks for the comment.
Reviewer 3 Report
The manuscript seemed to have improved compared to the original manuscript. But in my first review, I asked why a study protocol was not published prior to conducting this review. If not, should mention why prtocol was not published beforehand. Also, the authors provided search terms, but please add how many studies were retrieved from each search.
When discussing pain and comparions, outcome measures (VAS, AOFAS, daily activities, return to sports) should be mentioned and discussed because not all included studies used the same outcome measures. Also, not noly pain but FUNCTION should be discussed more in detail.
It would also be valuable if the authors included future research direction based on the preexisting studies.
Author Response
REVIEWER 3
Reviewer 3: The manuscript seemed to have improved compared to the original manuscript. But in my first review, I asked why a study protocol was not published prior to conducting this review. If not, should mention why prtocol was not published beforehand. Also, the authors provided search terms, but please add how many studies were retrieved from each search.
Authors: Dear Reviewer, thanks for the comment. We have not published the protocol due to the long response times of the registers. However, we modified the “limitations” section specifying that this is one on the limitations of our umbrella review, as you can read at lines 380. We added the number of results retrieved from each search as you requested as you can see in the new supplementary paper 1.
Reviewer 3: When discussing pain and comparions, outcome measures (VAS, AOFAS, daily activities, return to sports) should be mentioned and discussed because not all included studies used the same outcome measures. Also, not noly pain but FUNCTION should be discussed more in detail.
Authors: Dear Reviewer, thanks for the comment. We modified the manuscript as you suggested. We also added a specific section on “function” in the discussion, at lines 278-279 and 294-302.
Reviewer 3: It would also be valuable if the authors included future research direction based on the preexisting studies.
Authors: Dear Reviewer, thanks for the comment. We modified the manuscript as you suggested, as you can see at lines 373-377.